# Potential and challenges for an integrated management of tuberculosis, diabetes mellitus, and hypertension: A scoping review protocol

Vitri Widyaningsih[1], Ratih Puspita Febrinasari[1], Victoria Sari[2,3], Clarissa Augustania[2], Bintang Verlita[2], Chatarina Wahyuni[4], Bachti Alisjahbana[5], Ailiana Santosa[6], Nawi Ng[6,7], Ari Probandari[1] *

1 Faculty of Medicine, Universitas Sebelas Maret, Surakarta, Indonesia, 2 Disease Control Research Group, Universitas Sebelas Maret, Surakarta, Indonesia, 3 Master Program of Public Health, Universitas Sebelas Maret, Surakarta, Indonesia, 4 Faculty of Medicine, Universitas Airlangga, Surabaya, Indonesia, 5 Faculty of Medicine, Universitas Padjajaran, Bandung, Indonesia, 6 School of Public Health and Community Medicine, Institute of Medicine, Sahlgrenska Academy, University of Gothenburg, Gothenburg, Sweden, 7 Department of Epidemiology and Global Health, Faculty of Medicine, Umeå University, Umeå, Sweden

* ari.probandari@staff.uns.ac.id

**Data Availability Statement:** No datasets were generated or analysed during the current study. All

## Abstract

In many low- and middle-income countries (LMICs), the epidemiological transition is characterized by an increased burden of non-communicable diseases (NCDs) and the persistent challenge of infectious diseases. The transmission of tuberculosis, one of the leading infectious diseases, can be halted through active screening of risk groups and early case findings. Studies have reported comorbidities between tuberculosis (TB) and NCDs, which necessitates the development of an integrated disease management model. This scoping review discusses the possibilities and problems of integration in managing TB and NCDs, with a particular emphasis on diabetic mellitus (DM) and hypertension screening and control. We will conduct this review following Arksey and O'Malley's framework for scoping review. We will use key terms related to integrated management, i.e., screening, diagnosis, treatment, and care, of TB, DM, and hypertension in PubMed, Scopus Database, and ScienceDirect for research published from January 2005 to July 2021. This review will also consider grey literature, including unpublished literature and international disease management guidelines on TB, DM, and hypertension from WHO or other health professional organization. We will export the search results to citation manager software (EndNote). We will remove duplicates and apply the inclusion and exclusion criteria to identify the set of papers for the review. After screening the titles and abstract, two authors will independently review the full text of selected studies and extract the data. We will synthesize all selected studies qualitatively and the results will be discussed with the experts. The results will be used as the basis of the development of a guideline for integrated TB, DM, and hypertension management.

relevant data from this study will be made available upon study completion.

**Funding:** YES - AP, VW, CUW and BA received World Class Research Grant 2021-2022 by Ministry of Research and Technology/National Agency for Research and Innovation (RISTEK-BRIN), Republic of Indonesia (Grant No. B/112/E3/RA.00/2021). NN, AP, AS involvement is supported by the Swedish Research Council through the Swedish Research Link's Network Grant (Dnr: 2018-05194).

**Competing interests:** The authors have declared that no competing interests exist.

## Introduction

There has been a shift toward a more considerable burden of non-communicable diseases NCDs, particularly in low and middle-income countries (LMICs). In 2016, 71% of global death was related to NCDs [1], and 85% of premature mortality in LMICs was due to NCDs [2]. The two major contributors to NCD burdens are DM and hypertension. The prevalence of DM was 351.7 million in the working-age group (20–79 years) and 135.6 million among older people (65–99 years) [3]. Hypertension is the most common risk factor for death and disability, followed by smoking and high plasma glucose [1]. As the burden of NCDs increases, TB remains as a major cause of ill health and a leading cause of death, with an estimated 10 million new cases in 2019 [4].

The burden of TB is higher among patients with DM [5–7]. A meta-analysis in 16 countries reported that people with DM have a two to four-fold increased risk of TB [8], and TB prevalences among diabetic patients in developing Asian countries were 1.8–9.5 times higher than in the general population [9]. In 2013, approximately 15% of TB cases in adults were attributed to DM [10]. People with TB-DM comorbidities showed more severe TB presentations [11], increased risk of treatment failure, TB relapse, and delayed sputum culture conversion [12–15].

Several study also showed an association between TB and hypertension. Chung *et al.* reported a higher prevalence of hypertension (38.7% in the TB group vs 37.5% in the non-TB group) [16]. A study by Mandieka et al., also showed that people with latent TB have a higher risk of developing hypertension [17]. In a meta-analysis by Seegert et al., the prevalence of hypertension in TB patients ranges from 0.7% to 38.3%; however, there is no significant difference in hypertension cases between TB and non-TB patients in most case-control studies [18].

DM and hypertension also frequently occur together and worsen clinical outcomes [19, 20]. DM and hypertension share common risk factors and pathophysiology, resulting in the escalation of comorbidities, especially among low-income populations worldwide [21, 22]. Compared to people who only had DM or hypertension, people with DM and hypertension comorbidities had a higher rate of mortality [21] and are at increased risk for cardiovascular events [22]. Meanwhile, among TB patients, those with DM are at increased risk for cardiovascular diseases compared to those with only TB (24.5% vs 5.5%). Furthermore, TB patients with DM comorbidities are also more likely to use antihypertensive medication compared to patients with only a TB diagnosis (16.9% vs 3.2%) [23].

The relationship between TB and NCDs creates opportunities for the development of integrated management of these diseases [24–26]. Additionally, the management and control of chronic infectious diseases such as TB and NCD share many similarities. Both require long-term management and change in lifestyle, and most cases are being managed at the primary healthcare level [27]. Integrated management for TB, DM, and hypertension will be beneficial to improve the outcomes of these three diseases. Therefore, this scoping review will be carried out to explore the concept of health service integration in the management of TB, DM, and hypertension. In addition, this scoping review will seek to identify potential barriers and facilitators for integrated management of TB, DM, and hypertension.

## Materials and methods

The proposed scoping review will be conducted in accordance with the framework from Arksey and O'Malley: identifying the research question, identifying relevant studies, study selection, charting the data, collating, summarizing, reporting results, and conducting consultation [28]. To obtain a comprehensive view of integrated management of tuberculosis, hypertension, and diabetes, we will include original papers, grey literature including unpublished literature

and international disease management guidelines on TB, DM, and hypertension from WHO or other health professional organizations. We will include studies published from January 2005 to July 2021. The scoping review will be conducted from August 2021 to June 2022, starting with literature search, database creation and extraction.

## Stage 1: Identifying the research question

We formulated the research question as: "What is the potential for integrated management of TB, DM, or Hypertension?" and "What are the challenges for integrated management of TB, DM, or Hypertension?" Through this scoping review, we aim to present examples and potentials and challenges of integrating the management of TB, DM, and hypertension in any combinations i.e., TB and DM, TB and hypertension, and DM and hypertension. We will include activities related to screening, diagnosis, and treatment of the diseases.

## Stage 2: Identifying relevant studies

This scoping review includes the relevant studies based on the Joanna Briggs Institute (JBI) protocol [29]. Inclusion and exclusion criteria are presented in Table 1. Search strategy using keywords and queries can be found in Table 2. The literature search will be conducted in three databases: PubMed, Scopus Database, and ScienceDirect from January 2005 to July 2021. Hence, providing a review of literature for the past 15 years. Next, the title and abstract will be analyzed.

## Stage 3: Study selection

Two authors will screen the titles and abstracts of studies according to the selection criteria. In the scoping review, we will include all studies that evaluate the integration of disease management in screening, diagnosis, and treatment for TB and Hypertension, TB and DM, Hypertension and DM, and studies assessing the integration of the three diseases. We will exclude studies focusing on pregnant women with gestational diabetes and preeclampsia or studies in patients with immunodepressive symptoms who had a particular condition and usually require specific management. Two authors will independently review the full text of selected studies according to Fig 1. We resolved disagreements on study selection and data extraction by a discussion with one more reviewer if needed.

**Table 1. Inclusion and exclusion criteria.**

| Population | Concept | Context | Type of Sources |
|---|---|---|---|
| The target population included adult people with comorbidities of TB and or DM and or hypertension, i.e., people with TB and hypertension or DM, people with hypertension and DM, and people with the three diseases. Excluding papers which focus on specific patient subgroups, for example, children, pregnant women, and patients with immunodepressive symptoms. | Integrated TB, DM, hypertension management (screening, diagnosis, treatment or care). These include screening, diagnosis, and treatment of people with TB and hypertension or DM, hypertension and DM, and people with the three diseases. Excluding papers with a focus on single disease screening, diagnosis and treatment; | All health facilities (hospitals, health centers, and clinics), and intervention take place at communities. All types of indicators (input, process, output) of disease management, i.e. resources needed (human resources, funding, equipment, system), fidelity and effectiveness of existing integrated management implementation. We also include all indicators (patient, institution, health system).As this scoping review aims to explore potential and challenges for integrated management of these diseases, we plan to include all studies explaining the context of the disease management, with no specific exclusion criteria for context. | All types of original research; Papers with quantitative, qualitative or mixed-method study design; and papers written in English and published between the period of 2005–2021Excluding case report/ case series |

**Table 2. Keywords and queries for search startegy of integrated TB, DM, and hypertension management.**

| Database | Keywords and Queries |
|---|---|
| **Pubmed** | Keyword: 'Tuberculosis' (MeSH), "tuberculosis", "TB" 'Diabetes Mellitus' (MeSH), "diabetes mellitus", "DM". 'Hypertension' (MeSH), "hypertension", "high blood pressure", "integrated", "integration", "integrate", "management", "bidirectional screening", "screening", "diagnosis", "treatment", "care" Query: (((tuberculosis[MeSH Terms] OR tuberculosis OR TBC) AND (diabetes mellitus[MeSH Terms] OR "diabetes mellitus" OR DM) AND (hypertension[MeSH Terms] OR hypertension OR "high blood pressure")) OR ((tuberculosis[MeSH Terms] OR tuberculosis OR TBC) AND (diabetes mellitus[MeSH Terms] OR "diabetes mellitus" OR DM)) OR ((tuberculosis[MeSH Terms] OR tuberculosis OR TBC) AND (hypertension[MeSH Terms] OR hypertension OR "high blood pressure")) OR ((diabetes mellitus[MeSH Terms] OR "diabetes mellitus" OR DM) AND (hypertension[MeSH Terms] OR hypertension OR "high blood pressure"))) AND ((integrated OR integration OR integrate) AND (management OR "bidirectional screening" OR screening OR diagnosis OR treatment OR care)) |
| **Scopus** | Keyword: "tuberculosis", "TB", "diabetes mellitus", "DM". "hypertension", "high blood pressure", "integrated", "integration", "integrate", "management", "bidirectional screening", "screening", "diagnosis", "treatment", "care" Query: ALL ((tuberculosis OR TB) AND ("diabetes mellitus" OR DM) AND (hypertension OR "high blood pressure") AND (integrated OR integration OR integrate) AND (management OR "bidirectional screening" OR screening OR diagnosis OR treatment OR care)) |
| **ScienceDirect** | Keyword: "tuberculosis", "diabetes mellitus", "DM". "hypertension", "integration", "management", "bidirectional screening", "screening", "diagnosis", "treatment" Query: (tuberculosis) AND ("diabetes mellitus") AND (hypertension) AND (integrated OR integration OR integrate) AND (management OR "bidirectional screening" OR screening OR diagnosis OR treatment) |

## Stage 4: Charting the data

The two reviewers independently charted the data, discussed the results, and continuously updated the data-charting form in an iterative process. Data extraction will be carried out following the form that has been prepared in Table 3.

## Stage 5: Collating, summarizing, and reporting results

The findings obtained from the literature search from extracted data (Table 3) will be summarized in a table. Studies will be categorized by different characteristics, particularly on: study location (countries where the research was conducted: high and low middle-income countries), as well as types of integration (i.e., stage of integration and level of integration) (Table 3). A narrative report will be developed to map all the studies included in the scoping review. The summary of the findings will be presented in the different categories to present potentials and challenges for integrated management of the three diseases, particularly by the different types of integration. A qualitative synthesis will be conducted to present a robust summary of the literature, including barriers, facilitators, and recommendations on integrated diseases management. We will focus mainly on existing programs that have been implemented in the different stages of integration (screening, diagnosis, management) and levels of integration (primary health center, community, hospital). When possible, additional comparisons on whether there are differences between high and low-middle income countries will also be summarized and reported. Therefore, comprehensive information can be presented to address the potentials and challenges of integrated management of these diseases.

## Stage 6: Conducting consultation

The results will be consulted the experts and relevant stakeholders including regulators (health officers and national insurance officers), clinicians, and patients. This step is essential to validate the findings, receive feedback and obtain additional insights into the findings. Experts from India and Indonesia on TB, DM, and hypertension management will be invited for in-

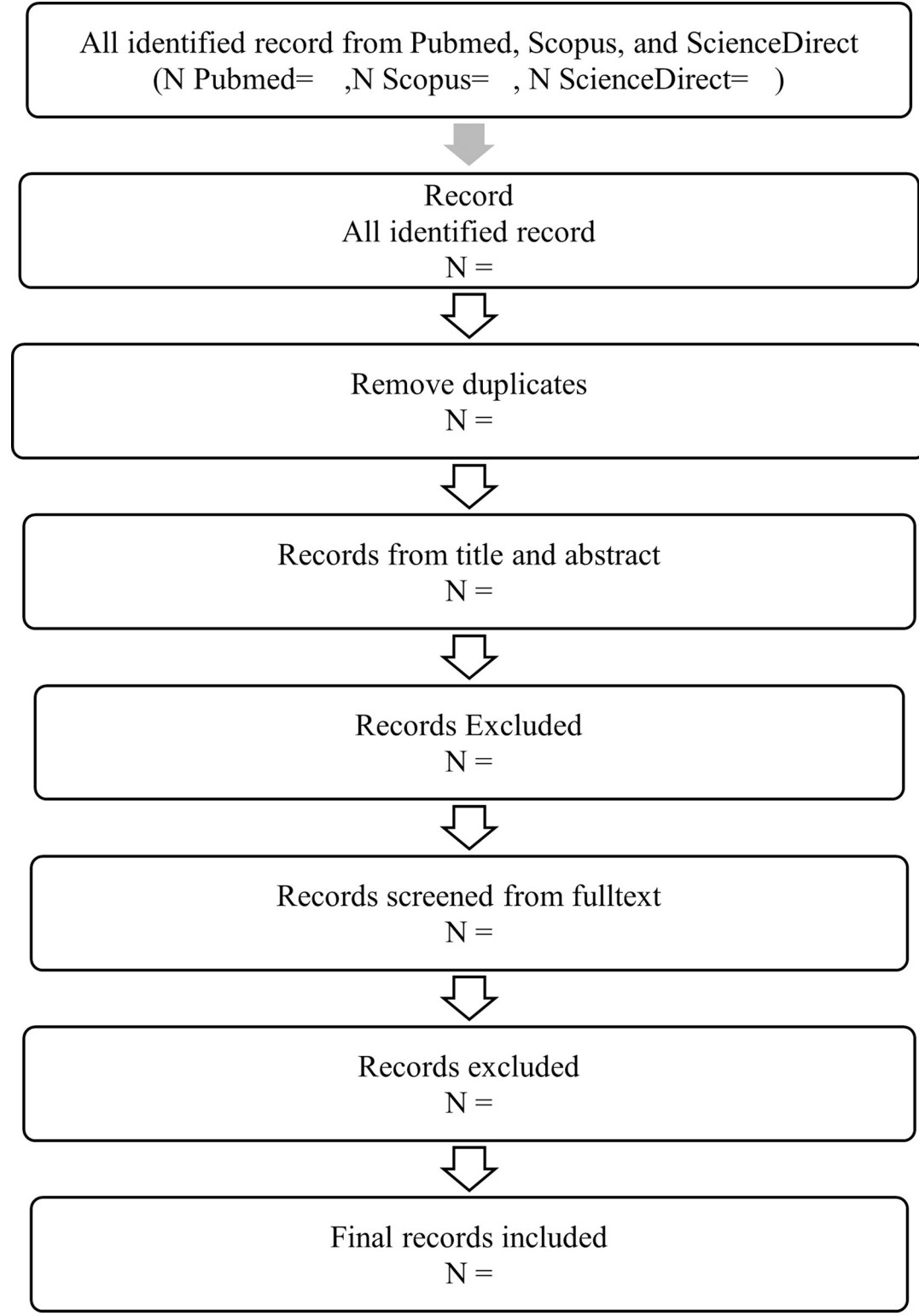

**Fig 1. Flow chart for the study selection.**

                   An integrated management of tuberculosis, diabetes mellitus, and hypertension: A scoping review protocol

**Table 3. Extraction data template.**

| Data | Data description |
|---|---|
| Article information | Author, year of publication, location of the study |
| Type of sources | A peer-reviewed journal, grey literature, and international guideline |
| Study Design | Quantitative, qualitative, mixed methods |
| Article Type | Original article, guidelines, unpublished reports |
| Aim | Overall aim or objective of the study |
| Population | The target population of the study |
| Research methods | Study design, sample size, methods of data collection, instruments used in the study |
| Concept of integration | Information on the stage of integration (screening, diagnosis, management) as well as activities conducted, level of integration (primary health center, community, hospital), disease focus (TB-DM, DM-hypertension, hypertension-TB, or TB, DM, and hypertension) |
| Results | Main findings of the study in the forms of quantitative results (%, mean/median/std deviation of some indicators, Odds Ratio, 95% CI). We will include information on code, themes, and meaningful quotes for further analyses. We will extract information on disease management guidelines for guidelines, particularly those relevant to the integrated disease management. |
| Additional information | Additional information of the study that relevant to the scoping review, for example study limitation and recommendation. |

depth interviews. India and Indonesia contribute to more than one-third of the TB burden globally [4]. Additionally, the two countries are also among the countries with the highest number of DM and hypertension [30, 31]. The selection of the experts will consider their research background and or program implementation experiences in TB, DM and hypertension management. After presenting the finding, the discussion will be centered on whether the findings are applicable in the context of healthcare in India and Indonesia and whether there are additional information on potential and barrier to integrated management of the diseases.

Meanwhile, the stakeholders' consultations will be conducted as focus group discussions, in which the findings will be presented and discussed. As for the stakeholder consultation, we plan to invite relevant stakeholders, mainly from Indonesia, to validate and provide feedback on our findings. For each group of stakeholders, a key person who can provide adequate information on the topic of integrated TB-DM and hypertension management will be identified. The stakeholder includes the Ministry of Health officials coordinating TB and NCD management, district health officials, and the officials from the national health insurance. A separate FGD with patients with TB and DM or hypertension comorbidities will be conducted. The patient FGD is conducted in several communities and hospitals in Indonesia's Central Java Province. This setting was because of the province's high prevalence of TB, DM, and hypertension. Central Java is also one of Indonesia's prioritized provinces for TB and NCD interventions.

## Ethical considerations

We have obtained ethical permits for this review from the Research Ethics Committee, Faculty of Medicine, Universitas Sebelas Maret, Indonesia, with the ethical review number: 09/UN27.06.6.1/KEP/EC/2021.

## Discussion

This scoping review will provide information on potential and challenges for integrated management of TB, DM, and hypertension, with different stages and levels of integration. A

systematic review on integrating TB and NCDs care in low-middle income countries (LMICs) highlighted the benefit of integrated disease management in improving health service delivery, mainly if comprehensive management from screening, referral, and treatment is conducted [32]. However, in this scoping review, we will focus on TB, DM and hypertension integration. Our scoping review will strengthen the arguments for implementing TB-DM integrated management. Increased case detection through bi-directional screening lead to more effective management of the diseases [33]. The integration reduced treatment loss to follow up for both TB-DM, increasing treatment success among TB patients [34]. However, synthesis of the integration among three diseases, i.e. TB, DM and hypertension, is still lacking.

The strengths of the scoping review include the inclusion of all types of health facilities (primary, secondary, and tertiary healthcare), consultation with experts, and the inclusion of clinical guidelines and grey literature. However, there are several potential limitations of the scoping review, including only literature published in English, and individual country clinical guidelines on TB, DM, and hypertension will not be included in this scoping review. Despite the limitation, to our knowledge, this is the first scoping review to map concepts relevant to the integrated management of TB, DM and hypertension. The results will be helpful for relevant stakeholders to improve the guideline for TB, DM and hypertension management in a high burden of the diseases. The findings will highlight research gaps in relevant to integrated disease management.

## Supporting information

**S1 File. PRISMA-ScR checklist.**
(PDF)

## Author Contributions

**Conceptualization:** Vitri Widyaningsih, Ratih Puspita Febrinasari, Chatarina Wahyuni, Bachti Alisjahbana, Ailiana Santosa, Nawi Ng, Ari Probandari.

**Data curation:** Vitri Widyaningsih, Ratih Puspita Febrinasari, Ari Probandari.

**Formal analysis:** Vitri Widyaningsih, Ratih Puspita Febrinasari, Ari Probandari.

**Funding acquisition:** Vitri Widyaningsih, Ari Probandari.

**Investigation:** Vitri Widyaningsih, Ratih Puspita Febrinasari, Victoria Sari, Clarissa Augustania, Bintang Verlita, Ari Probandari.

**Methodology:** Vitri Widyaningsih, Ratih Puspita Febrinasari, Chatarina Wahyuni, Bachti Alisjahbana, Ailiana Santosa, Nawi Ng, Ari Probandari.

**Project administration:** Vitri Widyaningsih, Ratih Puspita Febrinasari, Ari Probandari.

**Resources:** Vitri Widyaningsih, Ratih Puspita Febrinasari, Ari Probandari.

**Software:** Bintang Verlita.

**Supervision:** Ari Probandari.

**Validation:** Vitri Widyaningsih, Ratih Puspita Febrinasari, Ari Probandari.

**Visualization:** Victoria Sari, Clarissa Augustania, Bintang Verlita.

**Writing – original draft:** Vitri Widyaningsih.

**Writing – review & editing:** Vitri Widyaningsih, Ratih Puspita Febrinasari, Victoria Sari, Clarissa Augustania, Bintang Verlita, Chatarina Wahyuni, Bachti Alisjahbana, Ailiana Santosa, Nawi Ng, Ari Probandari.

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
