## [Decision Letter · Decision Letter 0]

5 Jan 2022

PONE-D-21-26078

Potential and challenges for an integrated management of Tuberculosis, Diabetes Mellitus, and Hypertension: a scoping review protocol

PLOS ONE

Dear Dr. Probandari,

Thank you for submitting your manuscript to PLOS ONE. After careful consideration, we feel that it has merit but does not fully meet PLOS ONE’s publication criteria as it currently stands. Therefore, we invite you to submit a revised version of the manuscript that addresses the points raised during the review process.

We have received the reports from our advisors on your manuscript, "Potential and challenges for an integrated management of Tuberculosis, Diabetes Mellitus, and Hypertension: a scoping review protocol", submitted to "PLOS ONE".

Based on the advice received, I feel that your manuscript could be reconsidered for publication should you be prepared to incorporate changes as suggested by reviewers. When preparing your revised manuscript, you are asked to carefully consider the reviewer comments which can be found below, and submit a list of responses to the comments. The final decision will be taken after your response letter and revision.

We look forward to receiving your revised manuscript.

Kind regards,

Masoud Foroutan, Ph.D; Assistant Professor

Academic Editor

PLOS ONE

Journal Requirements:

2. Please ensure that you refer to Figure 1 in your text as, if accepted, production will need this reference to link the reader to the figure.

3. Please upload a copy of Figure 2, to which you refer in your text on page 8. If the figure is no longer to be included as part of the submission please remove all reference to it within the text.

Additional Editor Comments (if provided):

Dear Dr. Probandari,

We have received the reports from our advisors on your manuscript, "Potential and challenges for an integrated management of Tuberculosis, Diabetes Mellitus, and Hypertension: a scoping review protocol", submitted to "PLOS ONE".

Based on the advice received, I feel that your manuscript could be reconsidered for publication should you be prepared to incorporate changes as suggested by reviewers. When preparing your revised manuscript, you are asked to carefully consider the reviewer comments which can be found below, and submit a list of responses to the comments. The final decision will be taken after your response letter and revision.

Reviewers' comments:

Reviewer's Responses to Questions

**Comments to the Author**

1. Does the manuscript provide a valid rationale for the proposed study, with clearly identified and justified research questions?

Reviewer #1: Partly

Reviewer #2: Yes

2. Is the protocol technically sound and planned in a manner that will lead to a meaningful outcome and allow testing the stated hypotheses?

Reviewer #1: Yes

Reviewer #2: Partly

3. Is the methodology feasible and described in sufficient detail to allow the work to be replicable?

Reviewer #1: Yes

Reviewer #2: No

4. Have the authors described where all data underlying the findings will be made available when the study is complete?

Reviewer #1: Yes

Reviewer #2: Yes

5. Is the manuscript presented in an intelligible fashion and written in standard English?

Reviewer #1: Yes

Reviewer #2: Yes

6. Review Comments to the Author

You may also provide optional suggestions and comments to authors that they might find helpful in planning their study.

Reviewer #1: 1. In line 148 (Stage 5): The author should explain in more detail about collating, summarizing, and reporting results.

2. In line 178 the parentheses have not closed.

3. In general, the discussion section is very short and concise, and it is better for the author to rewrite it in more detail.

Reviewer #2: General comments:

The introduction is well written and relevant citations are used. Overall, I think English needs improvement, authors use verb terms inconsistently. I would also suggest naming the sections of the protocol as per PLOS ONE requirements (https://journals.plos.org/plosone/s/what-we-publish/#loc-study-protocols) including the background, rationale, objectives, methodology, statistical plan, and organization of a research project. Content of the introduction section can be written under background and rationale; section, the identifying the research question, can be renamed as objective, all the rest can go under methodology and organization of the research project. I understand that a statistical plan is not relevant to your protocol.

Abstract

Line 33: Please specify key search terms for TB, DM, and Hypertension.

Line 36: Please specify in which citation manager software the results will be exported.

Introduction:

Lines 65, 66. Please mention the name of the author, as you do in the sentence before, otherwise it seems like that this finding is also from the same article (citation 16) and not from another article (citation 17).

Lines 71-72: Please provide a citation

Lines 74-75: Please mention: "… and DM and Cardiovascular events …", I understand that mortality rates were higher in both groups: Hypertension and DM and Cardiovascular events and DM vs Hypertension alone and Cardiovascular events alone

Lines 76-78: Not clear to me what do you mean. It is clear from previous citations (19-21) that DM is significantly associated with hypertension but citation 22 is interpreted so that TB can be an effect modifier among patients with DM. I believe that is not true. Could you please revise?

Materials and Methods

Lines 98-99: Why do you include studies published between January 2005 and July 2021, and not, for instance, including December 2021?

Lines 99-100: Have you already conducted the scoping review in Aug-Dec 2021 as it is mentioned in the protocol? If so, please ignore my comments above. If not, please revise the timeline.

Stage 1. Identifying the research question

Lines 103, 104: I would suggest splitting the research question into two research questions:

What is the potential for integrated management of Tuberculosis, Diabetes Mellitus, or Hypertension ?”

What are the challenges for integrated management of Tuberculosis, Diabetes Mellitus, or Hypertension ?”

Stage 2: identifying relevant studies

Line 118, Table 1: Could you please clarify whether the study population included patients only with TB or only with DM, or only with Hypertension? If so, how would you learn about integrating the management of TB with Hypertension, TB with DM, or DM and Hypertension in these groups? Also, is it your interest to review integrated management of TB and NCDs or are you also interested in reviewing integrated care of NCDs only, i.e., DM and Hypertension in your protocol?

The column of Context suggest specifying indicators for the program implementation process (input, process output, and outcome) and indicators at different levels of implementation (patient, institution, health system, region, district)

Would be good to see an annex with an exact list of inclusion and exclusion criteria.

Stage 3: study selection

Line 139: Figure1. Flow chart for the study selection is missing

Stage 4: Charting the data

Line 146 Table 4. Extraction Data Templates

Quantitative, Qualitative, Mixed Methods, etc. are study design examples and I suggest renaming the raw: Article type into study design or study type. Also, I suggest including research article, study protocols, Review, etc. in the separate raw named article type.

Stage 5: collating, summarizing, and reporting results

Line 151: specify what do you mean by study location, e.g., high income vs Low middle-income countries?

Stage 6: conducting consultation

Not sure what do you mean under the results will be consulted to the experts and relevant stakeholders. Do you mean focus group interviews with experts and relevant stakeholders? I understand that you plan to discuss the results and identify additional constructs and validate your findings from the article review? Please clarify how will you select experts and relevant stockholders? Will the discussion be structured facilitated as per the specially designed template? What questions do you plan to ask? How would you report the outcomes of the consultation?

Discussion

I suggest removing the discussion section. You can report about the limitations in your publication.

7. PLOS authors have the option to publish the peer review history of their article (what does this mean?). If published, this will include your full peer review and any attached files.

Reviewer #1: No

Reviewer #2: **Yes: **Veriko Mirtskhulava

---

## [Author Response · Author response to Decision Letter 0]

22 Feb 2022

Thank you for the detailed and comprehensive suggestions from the reviewers, that have improved the quality of our manuscript. We have carefully considered your comments.

Here, we explain how we revised the paper based on the comments and recommendations. We have made major changes as follow:

1. We elaborate the methods and discussion section to improve clarity of the scoping review we plan to conduct, particularly on the expert/stakeholder consultation. 

2. We provide point by point respond to the reviewers’ comments. 

Please find attached in this submission, our detail changes, and clarifications.

---

## [Decision Letter · Decision Letter 1]

19 Apr 2022

PONE-D-21-26078R1Potential and challenges for an integrated management of Tuberculosis, Diabetes Mellitus, and Hypertension: a scoping review protocolPLOS ONE

Dear Dr. Natalia Probandari,

Thank you for submitting your manuscript to PLOS ONE. After careful consideration, we feel that it has merit but does not fully meet PLOS ONE’s publication criteria as it currently stands. Therefore, we invite you to submit a revised version of the manuscript that addresses the points raised during the review process.

ACADEMIC EDITOR:Please submit your revised manuscript by Jun 03 2022 11:59PM. If you will need more time than this to complete your revisions, please reply to this message or contact the journal office at plosone@plos.org. Please include the following items when submitting your revised manuscript:A rebuttal letter that responds to each point raised by the academic editor and reviewer(s). You should upload this letter as a separate file labeled 'Response to Reviewers'.A marked-up copy of your manuscript that highlights changes made to the original version. You should upload this as a separate file labeled 'Revised Manuscript with Track Changes'.An unmarked version of your revised paper without tracked changes. You should upload this as a separate file labeled 'Manuscript'.

We look forward to receiving your revised manuscript.

Kind regards,

Masoud Foroutan, Ph.D; Assistant Professor

Academic Editor

PLOS ONE

Reviewers' comments:

Reviewer's Responses to Questions

**Comments to the Author**

1. Does the manuscript provide a valid rationale for the proposed study, with clearly identified and justified research questions?

Reviewer #2: Yes

2. Is the protocol technically sound and planned in a manner that will lead to a meaningful outcome and allow testing the stated hypotheses?

Reviewer #2: Partly

3. Is the methodology feasible and described in sufficient detail to allow the work to be replicable?

Reviewer #2: Yes

4. Have the authors described where all data underlying the findings will be made available when the study is complete?

Reviewer #2: No

5. Is the manuscript presented in an intelligible fashion and written in standard English?

Reviewer #2: No

6. Review Comments to the Author

You may also provide optional suggestions and comments to authors that they might find helpful in planning their study.

Reviewer #2: General comments:

1. To use abbreviations, you’ll first want to spell out the name or phrase, followed by the abbreviation in parentheses. Then, in any subsequent use of that name or phrase, only use the abbreviation.

Examples. In each case, you write out the full name and then introduce the abbreviation in parentheses: the word "tuberculosis (TB)" becomes the “TB” and the phrase “non-communicable disease (NCD)" becomes “NCD”.

2. Please make sure that the grammar and style used are for academic writing, please ask for assistance if needed.

3. Some of the sentences are in the past tense and some of them are in the future tense. My understanding is that since this is the study protocol, all sentences must be in the future tense, but I see that study according to this protocol has already been conducted. If that is so, I do not know what to recommend. Dear Editor, please provide guidance on that.

4. Study Protocols must also include the status and timeline of the study, including whether participant recruitment or data collection has begun where and when the data will be made available. See our Data Availability policy for more.

5. I would encourage you to remove the discussion section and use parts of it in the introduction and in the Materials and Methods.

6. Line 117, Table 1. Inclusion and exclusion criteria. The table is not clear enough. I have several questions/comments:

1. Did you include people with TB and hypertension?

2. Do you have age limitation for your study population?

3. I am not sure how this group (DM and hypertension) of the population addresses your research question; I understand your research questions are about the management of TB and NCDs (i.e., hypertension and DM); thus your population must have TB and at least one of the NCDs.

4. Do you exclude studies about people with TB only?

5. Do you exclude studies about all pregnant women, or the only the ones with gestational diabetes and/or preeclampsia/eclampsia?

Additional comments:

Abstract

Line 34-36, "This review will also 35 consider grey literature, including international disease management guidelines on tuberculosis, DM, and hypertension." Could you please specify what kind of gray literature other than clinical practice guidelines are you considering and what you mean by "international"? Do you mean WHO guidelines?

Lines 40-41, delete "will be synthesized"

Line 41, I suggest wording, the results will be used instead of could be used

Introduction

Line 63, please clarify what "previous study" you mean, as it is mentioned at the beginning of a new paragraph it is unclear what study you mean

Lines 95 -96, Could you please specify what kind of gray literature other than clinical practice guidelines are you considering and what you mean by "international"? Do you mean WHO guidelines? Do you consider the guidelines part of gray literature? I read from the text that you plan to review gray literature and international guidelines.

Stage 1: identifying the research question

Line 105, "… the management of any of hypertension, diabetes, and tuberculosis in any..", delete of any.

Line 112. correct spelling of "seatch"

Lines 114-115, "Hence, providing a review of literature for the past 15 years. Next, the title and abstract will be analyzed. " - something is missing, unclear to me, please clarify

7. PLOS authors have the option to publish the peer review history of their article (what does this mean?). If published, this will include your full peer review and any attached files.

Reviewer #2: **Yes: **Veriko Mirtskhulava

---

## [Author Response · Author response to Decision Letter 1]

6 May 2022

Thank you for your thorough review of our manuscript. We have revised and respond accordingly. Please find attached the revised version and the detailed description of the revisions. However, we have noticed that our first response to reviewer (RoR) is still available in addition to the second response to reviewer letter. We have tried several times to delete, but to no avail. Please refer to the RoR letter in the last section of the pdf

---

## [Decision Letter · Decision Letter 2]

29 Jun 2022

Potential and challenges for an integrated management of Tuberculosis, Diabetes Mellitus, and Hypertension: a scoping review protocol

PONE-D-21-26078R2

Dear Dr. Probandari,

We’re pleased to inform you that your manuscript has been judged scientifically suitable for publication and will be formally accepted for publication once it meets all outstanding technical requirements.

Kind regards,

Masoud Foroutan, Ph.D; Assistant Professor

Academic Editor

PLOS ONE

Additional Editor Comments (optional):

Reviewers' comments:

Reviewer's Responses to Questions

**Comments to the Author**

1. Does the manuscript provide a valid rationale for the proposed study, with clearly identified and justified research questions?

Reviewer #2: Yes

2. Is the protocol technically sound and planned in a manner that will lead to a meaningful outcome and allow testing the stated hypotheses?

Reviewer #2: Yes

3. Is the methodology feasible and described in sufficient detail to allow the work to be replicable?

Reviewer #2: Yes

4. Have the authors described where all data underlying the findings will be made available when the study is complete?

Reviewer #2: Yes

5. Is the manuscript presented in an intelligible fashion and written in standard English?

Reviewer #2: Yes

6. Review Comments to the Author

You may also provide optional suggestions and comments to authors that they might find helpful in planning their study.

Reviewer #2: I do not have any additional comments. Sorry for the delay in my response. It is an interesting study protocol.

7. PLOS authors have the option to publish the peer review history of their article (what does this mean?). If published, this will include your full peer review and any attached files.

Reviewer #2: **Yes: **Veriko Mirtskhulava

---

## [Editor Report · Acceptance letter]

4 Jul 2022

PONE-D-21-26078R2 

Potential and challenges for an integrated management of Tuberculosis, Diabetes Mellitus, and Hypertension: a scoping review protocol 

Dear Dr. Probandari:

I'm pleased to inform you that your manuscript has been deemed suitable for publication in PLOS ONE. Congratulations! Your manuscript is now with our production department. 

Kind regards, 

on behalf of

Dr. Masoud Foroutan 

Academic Editor

PLOS ONE